# Minority Stress and Positive Identity Aspects in Members of LGBTQ+ Parent Families: Literature Review and a Study Protocol for a Mixed-Methods Evidence Synthesis

**DOI:** 10.3390/children9091364

**Published:** 2022-09-07

**Authors:** Magdalena Siegel, Muriel Legler, Fortese Neziraj, Abbie E. Goldberg, Martina Zemp

**Affiliations:** 1Department of Clinical and Health Psychology, University of Vienna, 1010 Vienna, Austria; 2Department of Psychology, Clark University, Worcester, MA 01610-1477, USA

**Keywords:** LGBTQ+ parent families, sexual minorities, gender minorities, systematic review, minority stress

## Abstract

Background: Parents and children in LGBTQ+ parent families face unique stressors (i.e., minority stress), but also possess unique resources (i.e., positive identity aspects) related to their family identity. Empirical evidence and theory suggest that these minority stressors and positive identity aspects are situated on the individual, couple, and family level and may be associated with key outcomes, including parent and child health, family functioning, and school-related outcomes. A systematic evidence synthesis and a theoretical placement are currently lacking. The aims of the systematic review outlined in this protocol are thus to (1) map minority stressors and positive identity aspects according to multiple levels in the family system, and (2) to synthesize evidence on their associations with key outcomes. Methods: We will conduct a PRISMA-conform mixed-methods systematic review. Studies will be retrieved using a multi-tiered search strategy, including database searches (PsycInfo, PubMed, Scopus, Web of Science), cited literature searches, authors’ publication lists, and study requests. The mixed-methods synthesis will follow a parallel-results convergent synthesis design, where quantitative results will be synthesized via meta-analysis and qualitative results via thematic synthesis. Conclusions: Our proposed systematic review may add to the theoretical understanding of LGBTQ+ parent family functioning and advance social inclusion of LGBTQ+ parent families.

## 1. Introduction

LGBTQ+ parent families are families in which at least one parent identifies as lesbian, gay, bisexual, transgender, queer, or with another sexual orientation or gender identity than heterosexual or cisgender [1]. The umbrella acronym ‘LGBTQ+’ refers to lesbian, gay, bisexual, transgender, queer, and other sexual and gender minority identities. We use variants of this acronym when referring to specific subpopulations (e.g., ‘LG parents’ when referring to families with lesbian or gay parents). Parents and children in LGBTQ+ parent families face unique stressors (i.e., minority stress) but also possess unique resources (i.e., positive identity aspects) related to their collective identity as an LGBTQ+ parent family and the LGBTQ+ identity of individual family members [2].

Empirical evidence suggests that these stressors and resources have ramifications for parental and child mental and physical health [3,4,5,6,7,8,9], family relationships and functioning [10,11,12], and school-related outcomes [13,14,15]. Minority stress experiences and positive identity aspects are located on the individual, couple, and family level, and might also impact outcomes on these three levels in complex ways. Thus, it is important to understand the unique identity-based experiences of all members in LGBTQ+ parent families, and how they are related, to ensure LGBTQ+ parent families’ access to equal opportunities and thriving in society [2,13].

A systematic synthesis of this literature as well as a placement of this body of evidence within broader theoretical frameworks of LGBTQ+ health [16,17] and family functioning [18,19] are currently lacking. We aim to address these goals in the systematic review and meta-analysis outlined in the current protocol.

### 1.1. Minority Stress Experiences on the Individual Level: Conceptual Framework, Notable Extensions, and Empirical Evidence

Minority stress theory [17] provides the primary theoretical framework for describing the mental and physical health ramifications of (sexual and gender) identity-based stigma and discrimination, that LGBTQ+ individuals—including parents—face. Minority stress theory was originally developed to address unique stressors related to a sexual minority status, which are located on a distal (e.g., discrimination, harassment, victimization) to proximal (e.g., internalized negative attitudes towards one’s sexual orientation, concealment of the sexual orientation, expectation of rejection) continuum [17].

How minority stress impacts mental (and physical) health has been addressed on both the psychological and the neurobiological level. On the psychological level, minority stress processes are hypothesized as causing emotional dysregulation (e.g., rumination), social and interpersonal problems (e.g., social isolation), and maladaptive cognitive processes and schemas (e.g., hopelessness), which in turn lead to elevations in adverse (mental) health outcomes (e.g., depression, anxiety, suicidality) [20,21]. On the neurobiological level, minority stress exposure may cause neural adaptations that parallel those found in individuals with post-traumatic and other stress-related disorders [22]. Further, minority stress is conceptualized as disrupting the body’s adaptive stress response, causing physical health impairments related to systemic inflammation [23].

In the two decades since its inception, the minority stress model [17] has received further scrutiny. Several minority stressors such as sexual orientation concealment [24,25], rejection sensitivity [26], and internalized homonegativity (i.e., the internalization of negative societal attitudes towards non-heterosexuality [27]) have been theoretically and empirically refined, while new systemic stressors such as discriminatory laws and policies [28,29] have been discussed.

The minority stress model has been further extended to capture the experiences of transgender and gender-diverse (TGD) individuals [16,30,31]. Unique stressors related to a TGD identity include, for example, laws and policies failing to protect or even stigmatizing TGD individuals, gender-based victimization and rejection, being misgendered and not affirmed in one’s gender identity, the internalization of negative attitudes towards one’s gender identity (i.e., internalized transnegativity), or gender identity concealment [30,32]. These stressors are distinct from stressors related to sexual orientation, but their impact on mental and physical health is commonly described through the same psychological and neurobiological mechanisms [33].

Experiences of minority stress must be understood from an intersectional perspective [22,34,35,36]. This perspective acknowledges the unique and multifaceted lived experiences stemming from intersections of several marginalized as well as privileged identity dimensions, including sexual orientation, race, ethnicity, gender identity, class, or ability. Identity dimensions serve as moderators in the original, single-identity minority stress model [17], but recent works aim at addressing intersecting identity experiences directly, for example through in-depth, qualitative designs [37], innovations at the measurement level [38], or multi-dimensional frameworks grounded in minority stress theory [22,36].

Turning to empirical research on minority stress, research syntheses largely support the tenets of the minority stress model for LGBTQ+ individuals, particularly so for mental health outcomes such as suicidality [39,40], internalizing mental health problems [24,40,41], and substance use [42,43]. For physical health outcomes, evidence is more mixed [23,33], pointing to the role of further aspects related to stigmatization that are not captured by typical conceptualizations of minority stress (e.g., social safety [33]).

Research on associations between minority stress and the mental health of LGBTQ+ parents is scarcer. Well-documented mental health disparities within LGB+ populations [44] are, with exceptions [45], typically not found in LGB+ parent populations [46,47,48,49], with some studies even documenting better psychological adjustment [50,51]. The few studies that investigated associations between minority stress and mental health yielded mixed results: in studies with LG parents, internalized homonegativity was unrelated to anxiety [7,52] as well as to several indicators of physical health [5] when considered as a main effect and only inconsistently related to depression [5,7,52]. Further analyses revealed a moderating effect when considered with other minority stressors [7,52] and sociodemographic characteristics (e.g., gender [5]). Homophobic microaggressions in LG parents have been found to be related to psychological distress in bivariate analyses but not in higher-order models [4]. A global measure of minority stress has been linked to depressive symptoms in a sample of lesbian mothers [6]. For TGD parents, comparable evidence is currently lacking, but robust associations between minority stress and mental health outcomes have been found in TGD populations irrespective of parental status [32,40]. These findings underscore the importance of examining minority stress in LGBTQ+ parent populations specifically.

### 1.2. Individual- and Couple-Level Minority Stress in LGBTQ+ Individuals in Intimate Relationships

Minority stress, particularly with regard to sexual orientation, has also been theorized to impact the social functioning of LGBTQ+ people in general [20,21,53] and intimate relationship functioning in particular [54,55]. Thus, in LGBTQ+ parent families, minority stress might also affect the parental dyad.

Le Blanc et al.’s (2015) couple minority stress model outlines how (sexual) minority stress and intimate relationship functioning are linked through several reciprocal pathways [54]. This model introduces couple-level minority stressors that LGB(TQ+) individuals experience because of their societally marginalized sexuality and relationships. These stressors mirror those on the individual level and include processes such as legal discrimination (e.g., not being able to marry), relationship concealment, expectations of relationship rejection, or internalized negative evaluations of the relationship [54]. Individual-level (i.e., those described in the original minority stress model) and couple-level minority stressors, as well as minority stress differentials between partners, are assumed to be reciprocally associated with individual partners’ (mental) health and health behaviors as well as relational outcomes through individual and dyadic stress processes (see [54] for details).

Empirically, meta-analyses found small associations between individual-level minority stress and relationship quality in LGB+ individuals [56,57]. Couple-level minority stress has been found to contribute to adverse mental health outcomes in partnered LGBTQ+ individuals even after controlling for individual-level minority stress [58].

In LGBTQ+ parents specifically, evidence on associations between minority stress and relationship outcomes is limited: in a recent study with LGBTQ+ parents residing in the US, minority stressors such as low levels of outness, rejection expectations, and unequal rights were linked to lower relationship quality [11].

### 1.3. Individual-, Couple-, and Family-Level Minority Stress in Members of LGBTQ+ Parent Families

As outlined above, minority stress might (reciprocally) influence health-related and relational outcomes in LGBTQ+ parents through individual- and couple-level processes. We extend this notion to the family level—regarding both family-related outcomes and family-level stressors—by making three additional assumptions. First, minority stressors on the individual- and couple-level impact family-related outcomes (e.g., parenting, family relationships, family cohesion) directly or through other outcomes (e.g., through parental mental health). Second, LGBTQ+ parent families also face unique family-level minority stressors that impact key domains of functioning (e.g., mental health, parenting). Third, children in LGBTQ+ parent families may be impacted indirectly by parental minority stress (e.g., through parenting behaviors) as well as directly by their own (family-level) minority stress based on their identity as a child in an LGBTQ+ parent family.

A unifying framework for these notions, incorporating multiple family members, outcomes, and stressors, is currently lacking from a minority stress perspective (but see [2], for a resilience-focused perspective). However, a recent systematic review has formalized the impact of systemic forms of minority stress on parents and children in same-gender parent families (i.e., the legal vulnerability model [29]). While not focal to the model, several of its pathways describe associations between minority stress (on multiple levels) and family as well as child outcomes.

Regarding the first assumption, the model considers several family-level outcomes in LGBTQ+ parent families that could be impacted by minority stress either directly or indirectly: these include parental adjustment and parenting, family relationships, coparenting [18], and feelings of family legitimacy and cohesion [29]. Minority stress (experienced on the individual and, potentially, on the couple level) might impact these outcomes directly or indirectly, for example through impaired parental mental health or strained parental relationships (thereby limiting the parents’ self-efficacy in parenting or their ability to coparent [29]). Indeed, studies with LG parent families in Italy and the US document associations between proximal minority stressors (internalized homonegativity, rejection sensitivity) and more frequent conflicts about the child (a dimension of coparenting [10]) and higher parental stress [12]. In another study with LGBTQ+ parents, individual- and family-level minority stressors, such as low levels of outness and worries about family discrimination, predicted parental stress through lower levels of relationship satisfaction [11].

Regarding the second assumption, qualitative and quantitative research with LGBTQ+ parent families has documented unique family-level stressors, paralleling the notion of distinct couple-level stressors [54] described above. For example, an extension of the well-known minority stressor of internalized homonegativity on the family level might be the perceived burden to raise (or be) a child who is exceptional and well-adjusted (including being heterosexual and cisgender) in order to prove “normalcy” [59,60,61,62,63]. Another example, bearing parallels to rejection sensitivity [26], is the expectation and fear of rejection or discrimination of the family or the child in public settings [59,60,61,62,63,64,65]. Yet another proximal family-level stressor, paralleling sexual orientation or gender identity concealment on the individual level, is the concealment of the family structure in those settings [60,64,65,66]. LGBTQ+ parent families have also been found to be subjected to distinct distal forms of minority stress related to their family structure, such as microaggressions, discrimination, and stigmatization in various settings [9,59,67,68].

Regarding the third assumption, children in LGBTQ+ parent families might be directly impacted by minority stress they experience themselves as well as indirectly through minority stress that their parents experience. Child experiences of victimization are commonly investigated minority stressors in research on LGBTQ+ parent families. Importantly, when considered at the between-group level, children’s adjustment in LGBTQ+ parent families is similar to those raised in heterosexual, cisgender families [69,70], indicating that possible peer victimization experiences due to the family structure (or other minority stressors) are not driving adjustment disparities. However, within children in LGBTQ+ parent families, victimization and stigmatization have been found to be associated with higher levels of internalizing and externalizing problems [3,8,71,72,73,74,75], but not with substance use [76].

Studies linking parental minority stress to child adjustment are sparse and have yielded mixed results: some studies document negative associations between parental stigmatization or perceived heterosexism with child adjustment [73,77,78]. However, when considering several individual- and family-level minority stressors, only parental experiences of rejection were linked to child adjustment problems in a Dutch sample of lesbian mothers and their children [59].

### 1.4. Individual-, Couple-, and Family-Level Minority Stress in the School Setting

Associations between minority stress and children’s (as well as parents’) school and educational outcomes deserve special consideration but have received limited attention in research so far [13]. Academic achievement, similar to psychological adjustment, seems to be unrelated to parental sexual orientation or gender identity [70,79,80], suggesting that, at the between-group level, minority-stress-related experiences in the school setting are not detrimental to children’s educational adjustment. Still, at the within-group level, victimization experiences in the school have been negatively linked to child mental health [74], and also to school absenteeism [14].

Experiences of minority stress may also shape parents’ engagement in the school system. Parents, if resources allow, choose schools based on safety and diversity considerations for their children [14,81,82], but might still be confronted with disrespect, inexperience, or hetero- and cisnormative language regarding their family structure by school personnel [64,69,83,84]. There is some evidence that feelings of exclusion or experiences of discrimination might be associated with less school involvement by parents [15], who are, generally, very involved overall [14,15]. This, in turn, may impact children in LGBTQ+ parent families: for example, reduced parental school involvement was associated with more internalizing problems in a sample of preschoolers raised by LG (and heterosexual) adoptive parents [85].

### 1.5. Positive Identity Aspects in LGBTQ+ Individuals and LGBTQ+ Parent Families

The deficit-oriented perspective inherent to minority stress-based research reviewed above contributes little to our understanding of why LGBTQ+ individuals (including those in LGBTQ+ parent families) thrive in stigmatizing environments [2,86]. This is particularly important in the context of LGBTQ+ parent families, where both parents and children have been found to be psychologically well-adjusted despite ambiguous or even hostile social and legal contexts [9,29].

#### 1.5.1. Individual- and Couple-Level Positive Identity Aspects

A smaller body of literature, relatively disjoint from minority stress theory, has documented positive identity aspects related to an LGBTQ+ identity [31,87,88,89,90], irrespective of parental status: analogous to minority stressors, these aspects can be mapped to the individual and the couple level.

On the individual level, LGBTQ+ people attribute to their LGBTQ+ identity having increased empathy and compassion for others, being able to live authentically, enjoying freedom from societal norms and gender roles, taking pride in their identity and feeling connected to the LGBTQ+ community, valuing social justice, and engaging in activism [31,87,88,89,90]. On the couple level, having intimate relationships as an LGBTQ+ person has been reported to be central to gaining an increased capacity for intimacy and to be able to explore sexuality and relationship models [87,88,89,90].

Regarding empirical associations with individual and relational outcomes, these positive identity aspects have been linked to individual well-being (such as life satisfaction) in particular [91]. Evidence for associations with relational outcomes (such as relationship quality) is mixed [92,93].

#### 1.5.2. Family-Level Positive Identity Aspects

To the best of our knowledge, positive identity aspects in LGBTQ+ parent families have not been systematically studied in a similar fashion. However, by drawing on studies focusing on parents and children in LGBTQ+ parent families, we can assume analogous positive identity aspects on the family level (i.e., positive identity aspects because of being a member of an LGBTQ+ parent family): for example, lesbian mothers participating in the US National Longitudinal Lesbian Family Study, by now spanning three decades [94], have noted that they took pride in being role models and challenging prevailing family norms, drew strength from connections to the LGBTQ+ parent community, and enjoyed their freedom to parent across gender expectations [95]. Participants also described pride in raising respectful children that are appreciative of and sensitive to diversity [95]; a sentiment that has also been voiced in studies on bisexual and TGD parents’ family experiences [65,96,97]. Similarly, children raised in LGBTQ+ parent families describe being more open-minded, respectful, and accepting of diversity because of their family structure, enjoying freedom from gender norms and roles, and taking pride in engaging in LGBTQ+-related activism [61,98,99,100].

### 1.6. Objectives of the Current Review

Taken together, parents and children in LGBTQ+ parent families experience minority stressors and positive identity aspects on multiple levels. These experiences may be associated with parental health and child adjustment, family relationships and functioning, as well as school-related outcomes, through individual, shared, and reciprocal pathways within the family system. However, a comprehensive and systematic evidence synthesis is currently lacking. The objectives of our systematic review are thus two-fold:To map minority stressors and positive identity aspects experienced by parents and children in LGBTQ+ parent families according to multiple levels in the family system, specifically, the (i) individual, (ii) couple, and (iii) family level.To synthesize existing qualitative and quantitative evidence on associations between minority stress as well as positive identity aspects and key domains of LGBTQ+ parent families, namely (i) parental mental or physical health and child adjustment, (ii) family relationships and functioning, and (iii) school-related outcomes.

Our review will be guided by minority stress theory [16,17] and other LGBTQ+ specific theories and concepts, e.g., [87], as well as by general theories of family functioning, e.g., [18,19,101]. We will assume an intersectional perspective [22,35] that considers how the intersections of gender identity, race/ethnicity, sexual orientation, class, and other identity dimensions, as well as aspects related to the family form (e.g., route to parenthood) shape the experience of minority stress and positive identity aspects in all members of LGBTQ+ parent families. We will do so by addressing these intersections in our narrative syntheses and exploratory meta-regression models.

## 2. Materials and Methods

### 2.1. Adherence to PRISMA Guidelines and Open Science Practices

This protocol was prepared in line with the PRISMA [102], PRISMA-P [103], and PRISMA-S [104] guidelines for systematic reviews. The following sections are structured accordingly. Appendix A for this protocol are made available via the Open Science Framework (OSF; https://osf.io/4qd5k/ (accessed on 25 July 2022)), as will all study materials and datasets. The protocol will be registered in the International Prospective Register of Systematic Reviews (PROSPERO) after having undergone peer review.

### 2.2. Eligibility Criteria

We define our eligibility criteria following the PICOS framework (population, intervention/exposure, comparators, outcomes, study types) [105].

#### 2.2.1. Population

Study participants need to be members (i.e., parents or children) of LGBTQ+ parent families, defined as follows:

Definition of LGBTQ+ parent families: We define LGBTQ+ parent families as families consisting of two generations (i.e., the parent and child generation) in which at least one parent identifies as lesbian, gay, bisexual, transgender, queer, or otherwise as non-cisgender or nonheterosexual [1]. To account for the heterogeneity in the assessment of sexual orientation that we expect to encounter in primary studies, we will also include studies that assessed parental sexual orientation via other modes than self-reported identification. This includes the assessment of sexual orientation via attraction (i.e., people who report being attracted to their gender or people of any gender), behavior (i.e., people who report having had sexual experiences with people of their own gender or people of any gender), or (legal) relationship status (i.e., people who report being in or having had a relationship with a person of their own gender or any gender). While we refer to our study population using the identity-based term “LGBTQ+ parent families”, we consider identity, attraction, behavior, and relationships as separate, yet interrelated, aspects of sexual orientation [106]. Hence, we will clearly note and discuss the different modes of sexual orientation assessment used in primary studies and may—based on the characteristics of the final dataset—stratify our analyses accordingly. For the assessment of gender identity, we expect studies to primarily use self-report items but will include all forms of assessment.

We will include all forms of family and parenting constellations, e.g., planned families (via conception, adoption, or surrogacy), stepfamilies, blended families, collaborative-coparenting arrangements, or single-parent families. Studies on mixed-gender parent families will be included if at least one of the parents identifies as LGBTQ+. We will exclude studies focusing on grandparents (i.e., parents of parents in LGBTQ+ families) or other relatives of parents in LGBTQ+ parent families (e.g., siblings).

Parent generation: For this review, we consider a parent to be a person who takes on parenting responsibilities for a child, irrespective of biological or legal kinship [29].

Child generation: For this review, we consider a child in an LGBTQ+ parent family as an individual with at least one parent identifying as LGBTQ+ (as defined above). We will include cross-sectional or prospective studies with children under the age of 18 (sample mean) at the time of data collection (prospective studies: first measurement point). We will also include studies on children who are over the age of 18 (sample mean) at the time of data collection for studies that have a retrospective design that encompasses the time period in question (<18 years, e.g., [107]). We will exclude studies that focus on experiences during the pre-, perinatal or postpartum period (up to six weeks after birth based on clinical considerations [108]), given that recent systematic reviews have addressed these issues [68,109,110].

We will further exclude the following related types of studies based on the population investigated: studies focusing on (i) LGBTQ+ children or adolescents (unless they are “second generation” LGBTQ+, that is, at least one of their parents identifies as LGBTQ+ as well, e.g., [111]), (ii) mixed samples of families with minoritized identities (e.g., based on race/ethnicity or sexual orientation) without separate results focusing on LGBTQ+ parent families, (iii) childless couples or couples with unclear parental status (based on title and abstract information, e.g., [112]), and (iv) children with unclear parental LGBTQ+ status based on title and abstract.

#### 2.2.2. Intervention/Exposure

Based on theoretical considerations [17,20] that minority stressors and positive identity aspects cause variation in key outcomes, we consider minority stressors and positive identity aspects to be our intervention/exposure variables in the PICOS framework, even though we are synthesizing associations. Eligible quantitative studies thus need to include a measure of (a) minority stress and/or (b) positive identity aspects (see Table 1). Eligible qualitative studies need to mention either of these constructs in titles or abstracts, as this would indicate that it is a prominent theme within the study. Both families of constructs (a) and (b) are specified below.

Minority stress: We will include studies assessing aspects of minority stress on the individual, couple, and family level, as comprehensively reviewed in our introduction and summarized in Table 1. Importantly, distal forms of minority stress (e.g., discrimination or violence) do not need to be operationalized as being LGBTQ+-specific (e.g., experience of violence because of an LGBTQ+ identity).

Minority stress can be assessed either in the parent generation or the child generation and associated with outcomes assessed within the same person or in different members of the family. This means that we will, for example, include studies on associations between parental internalized homonegativity and child adjustment as well as on associations between parental internalized homonegativity and parental relationship satisfaction.

We will exclude measures of systemic and legal forms of minority stress (i.e., laws and policies, societal attitudes) because a recent systematic review [29] has comprehensively addressed this issue for same-gender parent families. We believe that the basic tenets of that synthesis (i.e., same-gender parent families are at an increased risk of experiencing adverse outcomes due to criminalization or lack of legal protection or recognition) hold for other forms of LGBTQ+ parent families as well. We will also exclude studies assessing general, minority-non-specific, stressors for (mental) health that have been discussed as risk factors in LGBTQ+ parent families [2], such as low self-esteem, family conflicts, or dysfunctional parenting not specific to the LGBTQ+ parent family status.

Positive identity aspects: Analogous to minority stress, we will include positive identity aspects located on the individual, couple, or family level as reviewed comprehensively in our introduction and in Table 1. We will exclude studies assessing general, minority-non-specific, protective factors for (mental) health that have been discussed as promoting resilience in LGBTQ+ parent families, such as general social support, self-esteem, or coping styles [2].

Of note is that the minority stressors and positive identity aspects outlined above and in Table 1 are not exhaustive. If we should encounter new or related constructs that, from their operationalization, fall within the theoretical frameworks outlined above, we will include these studies while noting our reasons transparently.

#### 2.2.3. Comparators

Not applicable (synthesis of within-group associations and qualitative data).

#### 2.2.4. Outcomes

We use an adapted typology of outcomes outlined in [29], as our review differs from that work in terms of the assessed intervention (i.e., systemic vs. distal/proximal forms of minority stress as well as positive identity aspects), but not in terms of assessed outcomes. Specifically, studies assessing one or more of the following outcomes are eligible for inclusion:

Parental mental or physical health and child adjustment (excepting school adjustment): For the parent generation, we will include any mental or physical health outcome, including measures of psychopathology (e.g., depressive symptomatology), general psychological distress, well-being (e.g., life satisfaction), markers of physical health (e.g., self-reported health), and substance use. For the child generation, we will include measures of externalizing (behavior) and internalizing (emotional) problems, attention problems, adjustment problems, somatic symptoms, and psychosocial adjustment, or any other measures of mental and physical health as defined for the parent generation.

Family relationships and family functioning: Measures of relationship quality (e.g., interparental relationship, parent–child relationship, sibling relationship), interparental or parent–child conflict, and parenting behaviors (e.g., coparenting, parental sensitivity) are included. We will not code outcomes focusing on relationships with people outside the family unit (e.g., relatives, friends).

School-related outcomes: We will include measures of educational attainment (child generation only) and school belonging and engagement (parent and child generation). Measures of educational attainment include, for example, school progress, grade-point average, or academic achievement. Measures of school belonging include feeling integrated into the school [113] and school adjustment (child generation only), school satisfaction, or school engagement (parent and child generation, e.g., [114]).

We do not place any restrictions on the mode of assessment or rater (e.g., parent, teacher, self). We will exclude outcomes related to financial well-being or parenthood aspirations.

#### 2.2.5. Study Type and Setting

We will include both published and unpublished literature written in English or German regardless of publication year or setting. Specifically, we will include empirical quantitative, qualitative, or mixed-methods studies that have been published in peer-reviewed journal articles or book chapters, or that are reported in research reports, dissertations, or other forms of grey literature. We will exclude non-empirical works (e.g., conceptual frameworks, commentaries), research syntheses (e.g., systematic reviews, meta-analyses), books, and Master’s theses.

### 2.3. Information Sources and Search Strategy

We will employ a multi-tiered search strategy to retrieve published and unpublished literature, based on database searches, cited literature searches, publication lists of relevant authors in the field, study requests posted to relevant listservs, and authors’ personal files.

#### 2.3.1. Database Searches

We will search the following databases: *PubMed*, *PsycInfo* (via EBSCOhost), *Scopus,* and *Web of Science* (Core Collection). No restrictions or database filters (e.g., coverage, population) will be set, except language filters (English, German). Search string development was formulated along two PICOS-eligibility criteria (population, intervention) and based on previously identified LGBTQ+ search terminology [115], systematic reviews or meta-analyses on LGBTQ+ parent families (see Appendix A for reference lists), terminology identified from seminal works on LGBTQ+ parent families [1,2,116,117], preliminary literature searches, and authors’ considerations. Further details and search strings adapted to all databases are reported in Appendix A.

Our search string is a combination of three sets of keywords relating to (1) sexual orientation and gender identity, (2) families, and (3) minority stress and positive identity aspects using the AND-operator (see Table 2). Note our population of interest is identified through a combination of sets (1) and (2). We deliberately opted against using a combined set of keywords for LGBTQ+ parent families (e.g., “same-gender parent families”, “lesbian mothers”), as we deemed the risk of not retrieving eligible studies (e.g., “children whose parents identify as gay or lesbian” [118]) as too high based on a preliminary literature search. We purposefully did not specify any keywords related to the PICOS-criteria outcomes or study types, as we deem the terminology to be too heterogeneous for inclusion in the search string and relevant records to be easily identified based on titles and abstracts.

Keywords from sets 1 and 2 will only be searched in record titles, whereas keywords from set 3 will be searched in record titles, abstracts, or keywords. This decision was made based on piloting the search string in *Scopus* (i.e., the database with the largest coverage), which yielded over 19,000 search hits when searching for keywords from all three sets in titles, abstracts, or keywords. We assume that most studies eligible for inclusion will mention the study population (and thus keywords from sets 1 and 2) in the title of their study.

The search string was developed jointly within the research team, iteratively refined by the first three authors [MS, ML, FN], subsequently adapted to all databases by the two second authors [ML, FN], and peer-reviewed [119] by the first author [MS]. We also validated our search string against a key sample of possibly eligible studies (*k* = 21; see Appendix A) in the database *Scopus*. Our systematic literature search will be updated prior to data synthesis.

#### 2.3.2. Cited Literature Searches

We will conduct cited and filtered literature searches of several key papers related to minority stress or positive identity aspects in LGBTQ+ populations. This search strategy will be employed to retrieve records not identified via our database search due to the heterogeneity in terminology relating to minority stress and positive identity aspects. Our rationale is that a study discussing/assessing forms of minority stress or positive identity aspects might cite the seminal works in this area when doing so, even when using different terminology from set (3) of our keywords.

We will conduct forward searches of the following key literature references in the database *Web of Science*, filtered for keywords from set (2) in record titles: [2,16,17,87,88,90]. In addition, we will conduct forward (*Web of Science*) and backward searches (reference lists) of all studies eligible for inclusion in our review.

#### 2.3.3. Publication Lists of Relevant Authors

We will hand-search publication lists (retrieved from personal websites, *Google Scholar*, or *Web of Science*) of the following authors known for their work on LGBTQ+ parent families: Katherine R. Allen, Jerry J. Bigner, Henny Bos, Victoria Clarke, Rachel Farr, Nanette Gartrell, Abbie E. Goldberg, Susan Golombok, Michael Lamb, Gary Mallon, Charlotte Patterson, and Corinne Reczek.

#### 2.3.4. Study Requests

We will post requests for unpublished data or studies to listservs relevant to LGBTQ+ and/or family psychology, including: APA Division 44 (Society for the Psychology of Sexual Orientation and Gender Diversity), ASA Section on the Sociology of Sexualities, European Public Health Association—Section LGBT, German Psychological Association—Section Family Psychology and Section Child and Youth Psychotherapy, International Academy of Family Psychology.

### 2.4. Study Records

#### 2.4.1. Data Management

Studies retrieved from database searches and cited literature searches using *Web of Science* will be deduplicated via *EndNote* following guidelines for systematic review deduplication [120]. Unique study records will be exported to an Excel spreadsheet (one row per unique record), including reference, title, and abstract information necessary for screening purposes. Records retrieved from other sources will be added to this file.

Full-text assessment and coding will also be carried out using Excel spreadsheets. Specifically, we will develop three standardized coding sheets based on our data items (see below): one sheet for (descriptive) study information (one row per study/record), one for the extraction of effect sizes (one row per effect size), and one for the extraction of results from qualitative studies (one row per reported impact of a minority stressor/a positive identity aspect on any outcome). The coding sheet for descriptive study information will be piloted and refined by all three coders [MS, ML, FN] on three sample records (at least one quantitative and one qualitative study) that were previously identified as eligible for inclusion.

For analyses and visualization, Excel spreadsheets will be imported into the open-source statistical software R, using the R Studio interface. Coding sheets will be made available upon publication as Excel sheets and native R-file formats (.rds).

#### 2.4.2. Selection Process

We will employ a standard two-stage selection process that consists of an initial title and abstract screening against the prespecified eligibility criteria followed by a full-text assessment. We will use a PRISMA flow chart [102] to visualize the study selection flow. No automation tools, crowdsourcing, or previously “known” assessments will be used in the selection process.

Title and abstract screening: For coder calibration, we will extract a random sample of 100 records. This sample will be screened independently by the first and second authors [MS, ML, FN] and open questions and ambiguities will be noted during the screening process and subsequently discussed within the team to ensure a shared understanding of the eligibility criteria and decisions. The first author [MS] will serve as a mentor during the screening process. In the next step, the remaining records will be screened independently by the two second authors [ML, FN]. Interrater reliability will be calculated for descriptive purposes using percentage agreement and adjusted Cohen κ [121] on these remaining records. All discrepancies will be resolved via discussion between the coders or arbitration by the first author [MS]. We will assume an overinclusive approach and include studies with unclear eligibility in the full-text assessment.

Full-text assessment: After full-text retrieval, the two second authors [ML, FN] will independently assess the obtained full texts against the eligibility criteria and extract information from eligible studies (see below). Eligibility decisions will be finalized before commencing data extraction. Again, all discrepancies will be resolved by discussion or arbitration [MS, MZ] and interrater reliability (percentage agreement, adjusted Cohen κ) will be calculated for descriptive purposes. Reasons for exclusion will be noted and made available alongside individual study references.

### 2.5. Data Collection Process

The two second authors [ML, FN] will independently extract information from eligible studies using standardized and piloted Excel sheets (see section “Data Management”). Similar to eligibility decisions, discrepancies will be resolved by discussion and arbitration [MS, MZ]. Interrater reliability will be calculated for all variables as percentage agreement and Cohen’s κ (categorical variables only). No automation tools will be used for data collection.

Extracted information from primary studies can be grouped into three categories: descriptive study information, effect sizes (i.e., results from quantitative studies or quantitative parts of mixed-methods studies), and qualitative results.

Extraction of descriptive study information: For extracted data items, see Appendix A. In case of missing information, we will code the respective entry as “not available”. In case of variables not relevant to the primary study, we will code the respective entry as “not applicable”.

Extraction of effect sizes: To synthesize quantitative evidence on the impact of minority stress and positive identity aspects on our specified outcomes, the primary effect size measures will be Pearson *r* (for presentation) and Fisher *z* (for analysis), which necessitates the extraction of the Pearson *r* point estimate as well as the effective sample size (for obtaining the sampling variance in the *z* metric). We will either (i) extract effect sizes based on these metrics directly (e.g., a correlation between parental mental health and parental internalized homonegativity), (ii) calculate them from relevant statistics (with or without calculation of intermediate standardized effect size metrics, e.g., from frequency tables or *t*-tests), or (iii) convert other standardized effect size measures (e.g., Cohen *d*) into this metric. For obtaining Pearson *r* from relevant statistics, we will use standard formulas [122,123] and transparently report in the published review which formulas were used for effect size calculation. For effect size conversion (e.g., conversion of Cohen *d* to Pearson *r*), we will follow the sequence as reported in [122].

If different effect sizes are reported for different subsamples (e.g., *r* = 0.25 for gay fathers and *r* = 0.30 for lesbian mothers) we will extract effect sizes on the subsample level, facilitating moderator analyses. We will not extract effect sizes pertaining to group differences between these groups (e.g., [4]).

Regarding the special case of regression coefficients, several approaches have been discussed in the literature (see [124]). We will follow the approach outlined by [125] as well as [126]. If a study reports standardized regression coefficients, we will first try to obtain the bivariate correlation between the two variables of interest from other tables, Appendix A, or the raw data. If a bivariate correlation cannot be extracted from the paper or its Appendix A and is not provided by the study authors (see below), we will attempt to extract semi-partial correlations and their sampling variances from reported model parameters (*t*-statistics of regression coefficients, *R*^2^) using formulas reported in [126]. We opted for the semi-partial correlation instead of the partial correlation, as this represents the more conservative estimate [127]. We will also note the number of covariates to be used as a predictor in a meta-regression model. These semi-partial correlations will be synthesized separately from bivariate correlations and not converted to Fisher *z* prior to main analyses [125].

Extraction of qualitative results: To synthesize qualitative evidence, we will code any instances in the results or discussion sections of primary studies that are concerned with the impact of one or more minority stressor(s) or one or more positive identity aspect(s) on any of the key outcome domains as defined above. Per instance, we will note (i) the specific stressor (e.g., discrimination) or positive identity aspect (e.g., connectedness to the LGBTQ+ parent community), (ii) the outcome (e.g., positive mood), (iii) the nature of the impact (verbatim, e.g., “I felt really good after the playdate [organized by an LGBTQ+ parent family organization], finally a place where I wasn’t the ‘lesbian mum’ on the playground” [fictional example]), and (iv) the person interviewed/providing the data (e.g., social parent). Subsequently, these statements will be synthesized using thematic synthesis [128], see section “Data Synthesis”.

Missing information from primary studies: As locating and contacting authors in case of any missing information (as defined by our data items below) in primary studies would exceed our resources, we will only inquire about missing information pertaining to sample sizes or effect sizes. We will send two reminders—two weeks and one month after initial contact—in case the authors do not reply to our inquiry.

### 2.6. Data Items

All data items alongside a preliminary coding scheme are reported in Appendix A. In short, we will extract the following information (adapted from [29]) from primary studies: (1) publication information (e.g., authors; publication type and status), (2) data collection information (e.g., type (qualitative/quantitative/mixed-methods) and timeframe of study; year, country, and mode of data collection), (3) sample information for parents and children (e.g., assessed generation; family type; sample size; sample composition regarding age, gender identity, sexual orientation, race/ethnicity), (4) information on minority stress/positive identity aspect assessed (e.g., type and rater), (5) outcome information (e.g., type and rater), (6) effect sizes (*r*, *N*), and (7) codes from qualitative data.

### 2.7. Outcomes and Prioritization

We consider all outcomes related to parental or child mental and physical health, family relationships and functioning, and school-related outcomes. We will not prioritize any outcomes in our synthesis nor are we planning to extract any secondary outcomes. We will group similar outcomes during the synthesis (e.g., outcomes concerned with mental health), but cannot preregister the exact nature at this point.

### 2.8. Risk of Bias in Individual Studies

We will assess the risk of bias in individual studies using the Quality Assessment with Diverse Studies (QuADS) tool [129]. The QuADS offers a set of 13 criteria to assess aspects of study quality across different designs and methodologies, which are evaluated on a narratively anchored four-point scale. We will follow procedures as described in the tool’s user guidelines [129], with the two second authors [ML, FN] serving as main coders. We will present results for descriptive purposes only, as we assume a rather exploratory, scoping approach in our review that is not primarily concerned with study quality.

### 2.9. Data Synthesis

#### 2.9.1. Effect Measures

Our effect size measure will be Pearson *r*, as our synthesis is concerned with associations between two variables (i.e., minority stress/positive identity aspect and an outcome). For analysis, point estimates expressed in Pearson *r* will be converted to Fisher *z* via the standard formula *z* = 0.5 * ln((1 + *r*)/(1 − *r*)) and sampling variances will be obtained using v_z_ = 1/(*n* − 3) [122]. For illustrative purposes, meta-analytic summary effect sizes and 95% confidence interval limits will be transformed back to Pearson *r*, using the standard formula *r* = (e^2*z^ − 1)/ (e^2*z^ + 1) [122]. For interpretation, we will adhere to well-established guidelines [130] and consider *r* = 0.10, 0.30, and 0.50 to be lower thresholds for small, medium, and large effects, respectively.

#### 2.9.2. Data Preparation

See section “Extraction of Effect Sizes”.

#### 2.9.3. Presentation of Results

Study characteristics of individual studies will be presented in tabular form, either in the main text or as a supplement. Other presentation formats we may employ are summary tables, meta-analytic forest plots (or variants thereof, e.g., a cumulative forest plot), or funnel plots (in case of assessing reporting biases) and will depend on the analyses conducted as well as their results.

#### 2.9.4. Synthesis of Effect Sizes

As we expect within-study dependency (e.g., several effect sizes on related constructs stemming from the same sample), we will conduct meta-analyses using robust variance estimation (RVE), which accounts for the dependency in the data structure by assuming a working model of the dependence structure [131]. As we currently do not know how our set of effect sizes will be structured in terms of populations (e.g., parent vs. child generation), interventions (e.g., types of minority stress), and outcomes, we regard preregistration of a specific working model (e.g., correlated and hierarchical effects, subgroup correlated effects) as premature. Rather, after the finalization of data extraction and grouping of effect sizes in conceptually meaningful categories, we will use the decision tree for selecting an appropriate working model as outlined in Figure 1 in [131]. In case we lack any empirical information about average correlations of effect sizes within samples, we will assume a correlation of *r* = 0.60 and conduct sensitivity analyses across a range of values from *r* = 0.00 to *r* = 0.90 (increase in 0.10 by step), noting changes in meta-analytic point estimates, standard errors, and τ^2^.

Based on the number of effect sizes, we may conduct exploratory single meta-regressions with the following potential moderators (final analyses based on conceptual considerations and number of effect sizes): generation (parent vs. child), family type (planned vs. stepfamily/blended vs. single parent), % lesbian/gay sexual orientation, % cisgender gender identity, % people of color. In the case of synthesis of semi-partial correlations (see section “Extraction of Effect Sizes”), we will conduct a meta-regression using the number of covariates as moderator [132].

We will use the restricted maximum-likelihood (REML) estimator to estimate the between-study variance due to its favorable performance in simulation studies and across a wide range of scenarios [131,133]. We will obtain Wald-type confidence intervals for the summary effect and profile likelihood confidence intervals for τ^2^. Additionally, we will use cluster wild bootstrapping for hypothesis tests in meta-regression models [134].

Apart from meta-analytic summary effect sizes and their 95% confidence intervals, we will report τ^2^ (with profile likelihood confidence intervals), *I*^2^, and results for the generalized/weighted least squares extension of Cochran’s Q-test for residual heterogeneity.

All analyses will be conducted in R, specifically using the {metafor} [135] and {clubsandwich} [136] packages for meta-analytic calculations and the meta-package {tidyverse} [137] for data manipulation and visualization. We will amend our protocol once data extraction has been completed and prior to analysis to preregister specifications of the working model for RVE as well as meta-regression models.

#### 2.9.5. Synthesis of Results from Qualitative Studies

Results of primary qualitative studies (or qualitative parts of mixed-methods studies) will be extracted and synthesized using thematic synthesis [128]. This synthesis technique has been used successfully in a previous systematic review on systemic forms of minority stress for LGBTQ+ parent families [29]. Thematic synthesis comprises three stages ranging from line-by-line coding over the organization of these codes into descriptive themes to the formation of analytical themes that go beyond the data [128]. Specifically, line-by-line coding will refer to the instances of minority stress/positive identity aspects and the perceived impact. From this, descriptive and consequently analytic themes will be formed in an iterative and consensual fashion. Three coders [MS, ML, FN] will be involved in the thematic synthesis.

For the development of the codebook used in our thematic synthesis, we will employ a purposive sampling approach (following [29]) to retrieve a pilot sample of eligible studies for initial code and theme development. Studies (approx. 5–10) will be selected for this pilot sample based on maximizing heterogeneity with regard to populations (children vs. parents), settings (US vs. other countries), interventions (positive identity aspects vs. minority stress on many levels), and outcomes. The first three authors will then independently perform line-by-line coding of these studies and develop an initial set of codes and descriptive themes. This initial codebook will be circulated among the research team and critically discussed with the senior author [MZ]. In the next step, the first three authors will independently code the remaining studies by employing a deductive-inductive approach. Thus, new codes and themes that may emerge will be incorporated if consensually deemed useful. Analytical themes will be formed after the coding of all studies has been completed, again first within coders and then in critical discussion with the senior author.

Regarding reflexivity [138], the three coders [MS, ML, FN] identify as cisgender with either a heterosexual or a diverse sexual orientation and do not have parenting experiences. Throughout the coding process, we will allocate time to engage in open discussions about the data and codes, aiming at elucidating the ways in which our perspectives in the analysis are influenced by our own experiences, perspectives, and values. In final discussions with the senior author, who has not been involved in study coding, our findings will be further validated. We consider the epistemological position of our systematic review to be positivist, with objectivity, rigorous systematization, and empiricism as hallmark features of this review type [139]. Our adherence to PRISMA guidelines, our preregistration approach, and replicable literature search, as well as our standardized coding materials and procedures, reflect this position.

#### 2.9.6. Overall Mixed-Methods Synthesis

For both research objectives, we will employ a parallel-results convergent synthesis design [140] for our mixed-methods synthesis. This means that we will synthesize results from quantitative and qualitative results separately (as described above) and provide a narrative discussion of how the findings from these two bodies of literature converge and/or complement each other in the discussion and recommendation sections [140]. For the first research objective, this might entail a tabular display of minority stressors and positive identity aspects, similar to Table 1, as well as a narrative discussion of how evidence from qualitative and quantitative studies differs or converges in this regard (e.g., some stressors might have emerged from qualitative studies, but have never been operationalized quantitatively). For the second research objective, this entails the presentation of findings derived from meta-analysis and thematic synthesis as described above, as well as a narrative summary of how these bodies of evidence differ or converge. In addition to minority stress and other LGBTQ+ specific theories as outlined in the introduction, we aim to integrate our findings within broader theories of family functioning [18,19,101]. Our narrative synthesis will be guided by an intersectional perspective [22,35] that considers how the intersections of gender identity, race/ethnicity, sexual orientation, class, and other identity dimensions shape the experience of minority stress and positive identity aspects in all members of LGBTQ+ parent families.

#### 2.9.7. Meta-Bias(es)/Reporting Bias Assessment

Applied meta-analysts are somewhat limited in assessing reporting bias in meta-analyses with dependent data. Only multilevel/RVE variants of Egger’s test [141] have been found to perform favorably (in terms of type 1 error rates) in simulation studies [142,143,144], and have been used in applied research syntheses [145]. Other families of methods, such as *p*-curve [146,147], *p*-uniform [148], or selection model approaches [149] are currently not suited to handle dependent effect sizes (and/or between-study heterogeneity) [143,144]. However, not only runs using only one method to detect reporting bias counter to recommendations favoring triangulation approaches [150,151] but the power of the multilevel/RVE variant of Egger’s test has also been found to be low [142,144].

Following recommendations [144], we will thus employ a mix of strategies to triangulate reporting bias and handle effect size dependency: First, we will conduct both multilevel and RVE variants of Egger’s test [144], thus incorporating dependency in reporting bias tests. Second, we will follow a more classical approach to handle effect size dependency by aggregating dependent effect sizes on an independent level [122,144]. This approach allows us to apply a range of standard reporting bias methods to a dataset consisting of independent effect sizes. Specifically, we will apply reporting bias methods from different families, including the trim-and-fill approach [152,153] (L0 estimator, side of imputed studies specified according to sign of summary effect), selection models [149], and the test of excess significance [154]. We will not employ *p*-value-based methods (*p*-curve, *p*-uniform) as simulation studies have shown unfavorable performance and they are not suited for handling between-study heterogeneity [143,148,155]. We will report adjusted summary effect sizes (excepting the test of excess significance), as well as dichotomous results of reporting bias tests (i.e., no bias present vs. bias present). In case our meta-analytic dataset consists of fewer than 10 independent effect sizes, we will report results descriptively but note the low power to detect reporting biases within this dataset.

We will conduct the reporting bias analyses on the same sets of studies we also use to compute summary effects. We will consider the following thresholds to indicate reporting bias: *p* < 0.10 for multilevel/RVE Egger’s test and the test of excess significance [141,154]; the difference between adjusted (mean values) and naïve meta-analytic summary effects exceeds 20% (either direction) in trim-and-fill and selection models [150].

#### 2.9.8. Confidence in Cumulative Evidence and Certainty of Evidence

We will discuss our confidence in the cumulative body of evidence narratively, drawing on topics outlined in the GRADE guidelines [156]. Specifically, we will discuss the results of risk of bias assessments and publication bias tests (if applicable). If a formal meta-analysis will be conducted, we will also include a discussion of summary effect sizes’ imprecision. Other GRADE criteria (e.g., indirectness, dose–response gradients) are more applicable to intervention studies and will not be discussed. For this reason, we will also not apply GRADE’s (numerical) certainty ratings, which do not apply to our presumably heterogeneous body of evidence (e.g., including qualitative studies).

#### 2.9.9. Timeline

Table 3 shows our proposed timeline for the conduction of the proposed systematic review, based on an average allocation of 20 h/week working time for the first three authors. We will commence working on the review (i.e., PROSPERO registration and literature retrieval) after provisional acceptance of the proposed protocol (expected in fall 2022).

## 3. Discussion

The importance of legal protection, social inclusion, and freedom from discrimination for LGBTQ+ parent families has been recognized by professional [157,158,159] as well as national and supranational organizations, such as the Healthy People 2020 initiative [160], the European Commission [161], the Council of Europe [162], and the United Nations [163,164]. Ensuring equality for LGBTQ+ parent families involves describing and mapping unique stressors and resources faced by all family members, as well as how they impact their well-being, family functioning, and educational attainment.

The proposed mixed-methods systematic review aims to achieve this by synthesizing qualitative and quantitative evidence and linking it to broader models of LGBTQ+ health (e.g., [17]) and family functioning (e.g., [18]). Its strengths will lie in the systematically gathered qualitative and quantitative evidence base gathered via a multi-tiered, comprehensive search strategy, its explicit grounding in theory allowing for the deduction of testable assumptions, as well as its sound methodology for both qualitative and quantitative evidence synthesis.

On the review level, some limitations will have to be discussed: First, the nature of our systematic literature search will likely yield a focus on legally progressive, mostly Western countries due to language and resource restrictions as well as authors’ cultural perspectives. Our findings will thus not generalize to LGBTQ+ parent families in other countries and cultural contexts, particularly those where parental sexual orientation or gender identity are criminalized or rendered institutionally invisible (e.g., [165]).

Second, studies on LGBTQ+ parent families have been conducted across different disciplines, display methodological plurality, and are grounded in various theoretical axioms [166,167]. Although we include various atheoretical search terms related to minority stress as well as positive identity aspects, our systematic search approach was predominantly informed by theories originating from the psychological literature [2,17] as well as scholars from the field of psychology. Thus, we might not be able to retrieve studies employing a very different terminology (e.g., from the field of sociology), particularly so for qualitative studies.

Third, we expect to find heterogeneous studies regarding investigated populations and operationalizations of stressors, positive identity aspects, and study outcomes, in addition to rather small sample sizes. This will likely limit our ability to meaningfully synthesize effect sizes through meta-analysis [168], as well as to assess moderator effects [169] or reporting biases [151].

Fourth, we will encounter limitations based on time and labor requirements, in addition to language restrictions as discussed above: for example, we will not be able to send inquiries about all missing data points relevant to our coding scheme to primary study authors. Instead, we decided to limit our inquiries to crucial information about sample and effect sizes. This may lead to the loss of potentially valuable information for readers, but should not bias main results (i.e., meta-analytic effect estimates). For the same reason (resource and time restrictions), we also did not extend our grey literature search to include websites from LGBTQ+ parent organizations or other web-based sources that might publish informal research reports. However, we believe that our multi-tiered search strategy (particularly forward-backward searches as well as inquiries via listservs) will be able to retrieve sufficient and representative grey literature in order to aid in our proposed mapping of minority stressors and positive identity aspects. Relatedly, the qualitative section of our mixed-methods evidence base would allow for a more in-depth qualitative synthesis (e.g., [170]) than our chosen approach of thematic synthesis [128]. This approach is in line with our study goal of systematically mapping minority stress and positive identity aspects to the individual, couple, and family levels. However, in the spirit of open science, we encourage researchers to extend our synthesis by engaging in a more nuanced re-analysis of our qualitative evidence base.

Possible limitations on the primary study level pertain to the sociodemographic characteristics of participants. Based on previous reviews [29,167], we expect study participants to be predominantly White, financially secure, middle class, cisgendered female, and living in same-gender relationships. While this may reflect the state of the field, it limits the generalizability of our review, particularly regarding parents and children of color, bisexual and queer parents, TGD parents, and families living in other forms of coparenting arrangements.

## 4. Conclusions

In all, our proposed systematic review has the potential to add to the theoretical understanding of family functioning in LGBTQ+ parent families [166,167] by elucidating the impact of minority-specific stressors and resources on key outcomes. This will aid in the conceptualization of future empirical studies and, more importantly, offer leverage points for advancing the social inclusion, legal recognition, and individual thriving of parents and children in LGBTQ+ parent families.

## Figures and Tables

**Table 1 children-09-01364-t001:** Non-exhaustive summary of eligible minority stressors and positive identity aspects on the individual, couple, and family level.

	Level		
Type	Individual	Couple	Family
Minority stressors	Distal: Harm to physical/sexual integrity (including victimization, harassment, physical or sexual abuse, hate crimes); unequal treatment/discrimination in various settings (e.g., work, healthcare); indignities (including insults, hate speech, name-calling, teasing, being misgendered, verbal harassment, microaggressions, micro-invalidations). Proximal: Sexual orientation/gender identity concealment (including outness [R], openness [R], disclosure [R]); internalized homo-, bi-, or transnegativity (including internalized heterosexism, internalized cissexism, internalized homo-, bi-, transphobia); rejection sensitivity (including expectations of rejection or fearing rejection).	Distal: Experiences of distal stressors as listed on the individual level, specifically because of being member(s) of an LGBTQ+ couple. Proximal: Concealment of the relationship or the partner (including avoidance of public display of affection); internalized negative attitudes towards LGBTQ+ relationships; rejection sensitivity regarding the relationship or partner (including expectations or fear of rejection).	Distal: Experiences of distal stressors as listed on the individual level, specifically because of being member(s) of an LGBTQ+ parent family. Proximal: Concealment of the family structure; internalized negative attitudes towards LGBTQ+ parent families or LGBTQ+ parenting (including feeling pressure to raise (or be) a child who is exceptional and well-adjusted); rejection sensitivity regarding the family or family members (e.g., children, including expectations and fearing discrimination of the child or other family members).
Positive identity aspects	Increased self-awareness due to LGBTQ+ identity; increased empathy and authenticity in social relationships due to LGBTQ+ identity; pride in the LGBTQ+ identity; feeling connected to the LGBTQ+ community; freedom from societal norms; serving as a role model; valuing social justice and engaging in activism.	Increased capacity for intimacy due to LGBTQ+ identity; exploration of sexuality and gender roles within relationships.	Serving as a role model for LGBTQ+ parents; feeling connected to the community of LGBTQ+ parent families; pride in raising/being a respectful child appreciative of diversity; freedom from gendered parenting roles; engaging in activism with regard to LGBTQ+ parent family issues.

Note. [R] = reversed. We consider minority stress and positive identity aspects experienced by children to map onto the family level, as their origins lie in the family structure. Distal minority stressors do not have to be operationalized as being based on LGBTQ+ (parent family) identity.

**Table 2 children-09-01364-t002:** Keywords used in the systematic literature search.

Block	Keywords
Sexual orientation and gender identity (Title)	bisexual * OR “female partnered wom?n” OR gay * OR “gender divers *” OR “gender identit *” OR “gender minorit *” OR “gender nonconforming” OR genderqueer * OR “gender transition *” OR GLB * OR homosexual* OR lesbian * OR LGB* OR “male partnered m?n” OR “men who have sex with men” OR “Non binary” OR nonbinary OR “non heterosexual” OR nonheterosexual* OR pansexual * OR plurisexual* OR queer * OR “same gender” OR “same sex” OR “sexual divers*” OR “sexual minorit *” OR “sexual orientation *” OR transfeminine OR transgender OR “trans m?n” OR transm?n OR transmasculine OR transpeople OR transperson OR transsexual OR “trans wom?n” OR transwom?n OR “two-spirit” OR “women who have sex with women”
Family (Title)	adopt * OR caregiv * OR child* OR cofamil * OR coparent * OR dad * OR famil * OR father * OR household * OR infant * OR interparent * OR marital OR marri * OR mom * OR mother * OR multiparent * OR mum* OR offspring * OR parent * OR polyfamil * OR remarriage OR stepdad * OR stepfamil * OR stepfather * OR stepmom * OR stepmother * OR stepmum * OR union *
Minority stress and positive identity aspects(Title/Abstract)	abuse OR antibisexual * OR antigay * OR antilesbian * OR antitransgender * OR binegativ * OR biphob * OR bully * OR “burden of proof” OR “community connectedness” OR conceal * OR disclosure OR discrimination * OR “expectation * of rejection” OR harass * OR heterosexis * OR homonegativ * OR homophob * OR marginalis * OR marginaliz * OR microaggression * OR microinvalidation * OR “minority identit *” OR “minority stress *” OR mistreat * OR openness OR outness OR “positive identit *” OR prejudic * OR pride OR queerphob * OR “rejection expectation *” OR “rejection sensitivity” OR resilienc* OR stigma * OR teas * OR transnegativ * OR transphob * OR “unequal treatment” OR victimization OR violen *

Note. * = truncation; ? = wildcard (e.g., “wom?n” maps to “woman” and “women”). For details and full search strings adapted to all databases, see Appendix A.

**Table 3 children-09-01364-t003:** Timeline for the systematic review.

	Month					
Work Package	1	2	3	4	5	6+
Preparation						
PROSPERO registration	X					
Literature Retrieval						
Database search, forward-search of key literature, and deduplication	X					
Search of publication lists from relevant authors	X					
Study requests	X					
Forward-backward-search of included studies				X		
Eligibility Check and Coding						
Screening based on titles and abstracts	X	X				
Full-text assessment and coding		X	X	X		
Analysis						
Qualitative synthesis				X	X	
Quantitative synthesis				X	X	
Write-Up						
Manuscript preparation					X	X

## Data Availability

All study materials can be found at https://osf.io/4qd5k/ (accessed on 25 July 2022).

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
