# Peer review of "Minority Stress and Positive Identity Aspects in Members of LGBTQ+ Parent Families: Literature Review and a Study Protocol for a Mixed-Methods Evidence Synthesis"

_children, 2022, doi:10.3390/children9091364_

Round 1

Reviewer 1 Report

This manuscript reviewed individual-, couple-, and family-level minority stressors facing LGBTQ+ parents and family. It also discussed positive identity aspects of the LGBTQ+ parent family. The review proposed a study protocol for the mixed-methods evidence synthesis. This manuscript is very informative, enlightening, and well-written. I appreciate the authors’ time and efforts on this project. I have some minor comments, which I elaborate on below.

1. The authors defined LGBTQ+ parent families as “families in which at least one parent identifies as lesbian, gay, bisexual, transgender, queer, or as otherwise non-heterosexual and/or non-cisgender” (line 32). I’d suggest the authors avoid using a majority group to define a minority group. Specifically, APA suggests not to use “non-heterosexual” or “non-cisgender” to define people with minority sexual orientation and gender identities. Thus, I recommend the author revise the definition to: “… queer, or other sexual orientation and gender identity than heterosexual and cisgender.”

2. In Table 1, the authors listed three levels of positive identity aspects. Yet, I couldn’t locate the discussion of individual-, couple-, and family-level positive identity aspects in the literature. The authors did a great job presenting the three levels of minority stress. Thus, I’d suggest the authors specify the three levels of positive identity aspects in their literature as well.

3. The authors discussed the potential limitations of the *results* of the review. However, what about the procedural and operational limitations, such as the time and labor requirements? 

4. I value this proposed systematic review. However, I hope the authors include a timeline or call-for-action in their conclusion. After reading this manuscript, I was unsure who is going to conduct this study: Do the authors plan to do this in the future, or do they encourage other researchers to do this? I’d like to know “what’s next” in the conclusion.

Author Response

We would like to thank the Reviewer for their positive feedback on
our manuscript. Please see the attachment for our point-by-point response.

Reviewer 2 Report

Summary -  I have read the manuscript with interest. The proposed study deals with a topic of high interest and relevance. The study is well conceptualized. Below are my suggestions for improvement. 

1. The introduction, while providing essential context for the study, is lengthy and can be edited for clarity. 

2. Please review use of "e.g." in the introduction statement and edit as appropriate. 

3. Reflexivity statement should be added for authors engaged in qualitative analysis. 

4. Please provide further details on how codebook for thematic analysis will be developed. Will the codebook be piloted on a subset of data for review? 

Author Response

We would like to thank the Reviewer for their encouraging feedback and suggestions, which have further strengthened our manuscript. Please see the attachment for our point-by-point response.
